# Homeschooling during COVID-19: Gender Differences in Work–Family Conflict and Alcohol Use Behaviour among Romantic Couples

**Danika I. DesRoches** [1,*], **S. Hélène Deacon** [1], **Lindsey M. Rodriguez** [2], **Simon B. Sherry** [1], **Raquel Nogueira-Arjona** [1], **Mariam M. Elgendi** [1], **Sandra Meier** [1], **Allan Abbass** [1], **Fiona E. King** [1] and **Sherry H. Stewart** [1]

1 Department of Psychology and Neuroscience, Dalhousie University, Halifax, NS B3H 4R2, Canada; Helene.Deacon@Dal.ca (S.H.D.); Simon.Sherry@Dal.ca (S.B.S.); rnarjona@gmail.com (R.N.-A.); M.Elgendi@dal.ca (M.M.E.); Sandra.Meier@iwk.nshealth.ca (S.M.); allan.abbass@dal.ca (A.A.); fiona.e.king.11@gmail.com (F.E.K.); Sherry.Stewart@Dal.Ca (S.H.S.)
2 Department of Psychology, University of South Florida, St. Petersburg, FL 33701, USA; lrodriguez12@usf.edu
* Correspondence: danika.desroches@dal.ca; Tel.: +1-506-962-6250

**Abstract:** Homeschooling due to COVID-19 school closures is likely to increase conflict between work and family demands, potentially leading to adverse substance-use effects. We conducted a survey with 758 couples focusing on homeschooling, work–family conflict, and alcohol use (April 2020). The 211 homeschooling couples reported more work–family conflict than the 547 non-homeschooling couples; there also were stronger effects on family interference with work in women. Among the homeschooling couples, homeschooling hours were associated with greater partner drinking. In distinguishable dyad analyses by gender, women's hours homeschooling were associated with greater drinking frequency by both parents. Men's hours homeschooling were associated with lower drinking frequency in their partners. Increased work–family conflict in homeschooling couples is particularly worrisome given its link to increased stress and poor mental health. Moreover, women's increased drinking may impede their ability to support their families during the pandemic. Men's increased drinking could put homeschooling mothers at risk for escalating conflict/domestic violence, given links of male drinking to intimate partner violence. Finally, the protective-partner effects of men's homeschooling hours on women's drinking frequency suggests that more egalitarian division of homeschooling labor may have protective cross-over effects.

**Keywords:** COVID-19; gender inequalities; homeschooling; work–family conflict; alcohol use

## 1. Introduction

In the spring of 2020, schools around the world closed in efforts to slow the spread of COVID-19 (UNESCO 2020). Consequently, many parents were obligated to homeschool their children while adjusting to changes in work arrangements (e.g., working from home, unemployment) and dealing with other pandemic-related stressors. This change forced many parents, particularly mothers, to adopt the role of teacher in addition to their roles as parents and as workers (Bariola and Collins 2021). Theory predicts that holding multiple roles can exacerbate conflict between work and family, and evidence shows that increased work–family conflict (WFC) is linked to alcohol use, especially in women (e.g., Roos et al. 2006). Accordingly, the current study aimed to examine the impacts of COVID-19 mandated homeschooling on WFC and alcohol use in romantic couples and their possible moderation by gender.

The psychological consequences of the pandemic are predicted to be especially negative for those with children at home (e.g., Brooks et al. 2020; Wenham et al. 2020). Indeed, recent data seem to support this prediction. In May 2020, Gadermann et al. (2021) surveyed more than 3000 Canadians and showed parents were more likely to report worse

mental health since the onset of the COVID-19 pandemic than nonparents. These adverse consequences may be amplified for parents who take on the additional responsibility of homeschooling. A daily diary study during the pandemic found that parents reported lower wellbeing on days they needed to homeschool their children (Schmidt et al. 2020).

Parents who were required to homeschool their children during the pandemic were left juggling the new role of teacher on top of their increased parenting and work demands. More than 7 out 10 of parents cited COVID-19 mandated homeschooling to be a significant source of stress (APA 2020). Further, parents described mandated homeschooling during the pandemic as a "difficult situation" and as a responsibility that was beyond their personal capability (Parczewska 2020). Though mothers and fathers were equally likely to describe homeschooling as a "difficult situation", mothers were more likely to rate it as being beyond their capabilities (Parczewska 2020).

Pre-pandemic research with families who are homeschooling on a voluntary basis supports the idea that homeschooling can be burdensome for the parent fulfilling the role of "teacher". A recent review of the voluntary homeschooling literature led authors to argue that taking on the additional role of teacher can worsen mental health issues (Baker 2019). For instance, in a qualitative study looking at the experiences of mothers who decided to homeschool, it was commonly reported that the teacher role quickly became overwhelming (Lois 2006). Many mothers reported experiencing conflicts between their roles as teachers and as parents, leading some to burn out (Lois 2006). Despite these consequences for parents, prior to the pandemic many families still chose to homeschool their children for reasons such as religious ideology, health problems, a lack of faith in the education system and concerns about bullying (Anthony and Burroughs 2010; Dumas et al. 2010). Contrastingly, during COVID-19 many families were stripped of the capacity to make these decisions and were mandated to homeschool their children, which intuitively might have more detrimental effects on their mental health and substance use. The multiple-burdens hypothesis posits that holding too many roles can lead to increased levels of WFC (Kuntsche et al. 2009). WFC arises when competing demands from work and family domains are mutually incompatible (Greenhaus and Beutell 1985) and can emerge when family interferes with work (FIW) and/or when work interferes with family (WIF) (Netemeyer et al. 1996). Pre-pandemic research suggests that, consistent with traditional gender roles, women experience more FIW conflicts (Fu and Shaffer 2001; Keene and Quadagno 2004) while men experience more WIF conflicts (Fu and Shaffer 2001; Kulik et al. 2016).

To examine the impact of COVID-19 on WFC, Schieman et al. (2021) asked 1843 Canadian workers about their WFC levels before (September 2019) and after (April and June 2020) the onset of the pandemic. Workers who were not living with children or who were living with a teenager showed a decrease in WFC after the pandemic began. This could be due to COVID-19 restrictions (e.g., social distancing) limiting the "life" part of the work–life conflict equation. In contrast, no decrease in WFC was observed among workers living with children under 13 years old. Interestingly, workers living with children between the ages of 6 and 12 reported the highest levels of WFC, followed by those living with children under 6. Given the nationwide school closures in place in April through June 2020 (UNESCO 2020), Canadian parents of school-aged children (6 to 12 years of age) would have experienced the added demand of COVID-19 mandated homeschooling during this time. Taking on the additional role of teacher may have created additional role burdens for working parents and, consequently, less opportunity for reduced levels of WFC during the pandemic.

Research shows the responsibility of homeschooling during the pandemic has largely fallen to women (Del Boca et al. 2020; Shafer et al. 2020). In a study of 2200 Americans, 80% of women reported that they spend more time on homeschooling than their partners (Miller 2020). Evidence suggests family demands are a more important predictor of WFC for women than for men (Elliott 2008; McElwain et al. 2005). Therefore, the added demand of COVID-19 mandated homeschooling might be a risk factor for increased WFC, particu-

larly among women. Indeed, women reported significantly more WFC than men during the COVID-19 pandemic lockdown in Spain during the Spring of 2020, when schools in the country were closed (López-Núñez et al. 2021).

There is extensive evidence that links conflicts between work and family demands to greater alcohol use (Roos et al. 2006; Wang et al. 2010; Wolff et al. 2013). Pre-pandemic findings suggest stronger links of WFC to increased alcohol use in women than men (Kuntsche and Kuntsche 2021; Kuntsche et al. 2009; Roos et al. 2006). Consistent with the multiple-burdens hypothesis, the risk for heavy drinking is higher for women who hold three roles (i.e., partner, parent, and paid worker) versus one role; such effects of multiple roles are not evident for men's drinking (Kuntsche et al. 2009). Similarly, WFC has been found to be associated with heavy drinking in women but not in men (Roos et al. 2006).

Further, residing with children during the COVID-19 pandemic has proven to be a risk factor for increased self-reported alcohol use (Gadermann et al. 2021; Rodriguez et al. 2020; Schmits and Glowacz 2021). Among parents, increased alcohol consumption during COVID-19 was more prevalent among men than women (Gadermann et al. 2021). However, pandemic-related psychological distress was found to be more strongly related to increases in drinking among women than men (Rodriguez et al. 2020).

It is not yet clear how the roles and the responsibilities added by COVID-19 mandated homeschooling play out in couples. Certainly, in the couples' stress and coping literature, an individual's stress levels affect not only their own but also their partner's coping behavior (O'Brien and DeLongis 1997). That is, adverse effects of stress can cross over from one partner to another (Neff and Karney 2007). A recent study showed the stress experienced by the parent providing the COVID-19 mandated homeschooling had cross-over effects on the coping-related alcohol use of their romantic partner (Deacon et al. forthcoming). Given that coping motives predict heavier and more problematic drinking (Cooper et al. 2016), it remains important to examine whether there are similar effects of mandatory homeschooling on alcohol-use behavior and to examine what impacts of homeschooling (e.g., effects on WFC) might underlie such effects on drinking.

In the present couples' study, we examined mandatory homeschooling during the pandemic as a potential risk factor for increased WFC and alcohol use, including testing if these effects are different for women and men. We also examined if cross-over effects of homeschooling exist in couples (Neff and Karney 2007), and whether these vary by gender. We first examined homeschooling effects by comparing couples who were and were not homeschooling. Recognizing involvement in homeschooling may vary across members of a couple (Deacon et al. forthcoming), our second set of analyses examined our outcomes as a function of each member's time spent homeschooling. Knowledge about the impact of mandated homeschooling on women's and men's WFC and alcohol use is needed for planning responses to future waves of COVID-19 and its variants as well as other predicted future pandemics (Parczewska 2020). Further, this information could help develop strategies to mitigate the observed impact of the pandemic on gender inequality (Dang and Nguyen 2021; Reichelt et al. 2021).

Based on previous research (e.g., Fu and Shaffer 2001; Kulik et al. 2016), we expected the following:

**Hypothesis 1.** *Women would experience more FIW conflicts than men and men would experience more WIF conflicts than women.*

**Hypothesis 2.** *Consistent with the multiple-burdens hypothesis, homeschooling couples would show higher levels of WFC and alcohol use than non-homeschooling couples, with stronger effects of homeschooling status on WFC and drinking in women than in men.*

**Hypothesis 3.** *Among homeschooling couples, women would spend more time homeschooling than men.*

Time spent homeschooling would increase:

**Hypothesis 4.** *Own WFC and drinking (actor effects).*

**Hypothesis 5.** *Particularly in women.*

**Hypothesis 6.** *Cross-over effects on their partner's outcomes (partner effects).*

## 2. Materials and Methods

### 2.1. Participants and Procedure

Participants were 758 romantic couples (1516 individuals; *M* age = 54.7 years, *SD* = 13.9) living in Canada. Sociodemographic information and relationship characteristics are in Table 1. Couples reported being in a romantic relationship for an average length of 27.02 years (*SD* = 0.38) (range = 1.08 to 64.83 years). Of the total sample, 211 couples reported homeschooling during April 2020, with the majority (86% (n = 173 couples)]) doing so because of COVID-19. These couples reported, on average, homeschooling 2.0 children (*SD* = 0.96), with the average age of homeschooled children being 9.84 years (*SD* = 4.86). We looked at whether participants who were homeschooling differed from participants who were not homeschooling on demographic variables (Table 1). Homeschooling couples reported being together for significantly less time (*M* = 224.2 months; *SD* = 106.2) than non-homeschooling couples (*M* = 362.80 months; *SD* = 183.90), $t(1292) = 18.26$, $p < 0.001$. Further, participants who were homeschooling were more likely to be college/university graduates, and have some post-graduate education or a post graduate degree compared to participants who were not homeschooling, $X^2(1, N = 1423) = 57.15$, $p < 0.001$. Participants who were homeschooling were also more likely than participants who were not homeschooling to be White, $X^2(1, N = 1516) = 87.23$, $p < 0.001$, and to be working full-time, $X^2(1, N = 1407) = 139.78$, $p < 0.001$. However, participants who were homeschooling were not more or less likely to be working part-time compared to participants who were not homeschooling, $X^2(1, N = 1407) = 1.89$, $p = 0.169$.

The nature of COVID-19 mandated homeschooling differed across Canadian provinces. While some provincial governments (e.g., Alberta, Ontario) provided requirements for the amount or type of schoolwork expected weekly, others (e.g., British Columbia, Quebec) left each school district to develop learning plans (Li 2020) and offered access to websites (e.g., Keep learning BC, Open School) and other resources (e.g., take-home learning packages). Schools and individual teachers also added to the variability. Our cross-national Canada-wide recruitment captured this diversity.

Couples completed the online survey through Qualtrics Panel Surveys in early July 2020 and retrospectively reported on the month of April 2020. This period was chosen because schools across Canada were closed, and strict containment measures were in place, forcing couples to spend most of their time at home together. Participants provided informed consent and were screened for eligibility. Eligible panelists were involved in a romantic relationship of at least 3 months' duration with a partner who was also willing and available to participate. Both partners had to be at least 19 years old, living in Canada, cohabiting together during April 2020, and following COVID-19-related stay-at-home advisories in their jurisdiction during April 2020. Essential workers were excluded as we wanted to include only couples who were spending most of their time together at home and who would have had the opportunity to provide homeschooling to their children. This study was approved by a university research ethics board (#2020-5166).

**Table 1.** Demographics information and relationship characteristics for homeschooling and non-homeschooling romantic couples living in Canada.

| Variable | Non-Homeschooling (*N* = 1094) | Homeschooling (*N* = 422) |
|---|---|---|
| Relationship Length (in years)—*M*(*SD*) | 30.24(15.32) | 18.68(8.85) |
| **Gender** | | |
| Female | 48.18% | 50.23% |
| Male | 51.55% | 49.53% |
| Non-binary/Unknown | 0.27% | 0.24% |
| **Relationship Status** | | |
| Mixed Sex | 93.60% | 94.76% |
| Same sex | 6.40% | 5.24% |
| **Marital Status** | | |
| Married/Common Law | 98.91% | 99.05% |
| In a serious relationship | 1.09% | 0.95% |
| **Employment Status** | | |
| Employed full-time | 26.60% | 59.00% |
| Employed part-time | 10.24% | 8.06% |
| Unemployed/Students | 55.85% | 27.01% |
| Unknown | 7.31% | 5.93% |
| **Highest Level of Education** | | |
| Elementary school | 0.91% | - |
| Some high school | 4.19% | 0.95% |
| High school graduate | 17.85% | 10.67% |
| Some college/university | 18.12% | 7.82% |
| College/university graduate | 43.08% | 52.13% |
| Some post-graduate | 4.19% | 5.45% |
| Post-graduate degree (e.g., Master's, Ph.D., LLB, MD) | 11.39% | 22.98% |
| Unknown | 0.27% | - |
| **Province** | | |
| Ontario | 44.44% | 51.19% |
| Alberta | 12.94% | 17.06% |
| British Columbia | 16.40% | 11.85% |
| Quebec | 7.28% | 8.54% |
| Atlantic Provinces [1] | 10.56% | 6.62% |
| Manitoba | 4.19% | 2.84% |
| Saskatchewan | 4.19% | 1.42% |
| North West Territories | - | 0.48% |
| **Ethnicity** | | |
| White | 82.08% | 59.95% |
| Asian or Arab/West Asian (e.g., Armenian, Egyptian, Iranian, Lebanese, Moroccan) [2] | 11.97% | 30.57% |
| Latin America or Black or First Nations [2] | 1.26% | 4.27% |
| Multiracial | 2.19% | 1.66% |
| Other/Unknown | 2.56% | 3.55% |

[1] Combined to maintain confidentiality of respondents due to low numbers in one or more of these provinces. [2] Combined to maintain confidentiality of respondents due to low numbers in one or more of these categories.

### 2.2. Measures

Both members of the couple completed the same set of measures at the beginning of July 2020 using the same 1-month timeframe (i.e., April 2020) as the reference for measure completion.

### 2.2.1. Homeschooling Assessment

Participants were first asked if they or their partner had homeschooled any children in Grades 1–12 (i.e., homeschooling status; 0 = no, 1 = yes). If so, they were also asked about

the time they had spent homeschooling in hours per week (Guterman and Neuman 2018, 2020).

### 2.2.2. Demographics

We assessed several demographic and relationship variables (e.g., gender, age, race, relationship status, length of relationship) as well as whether participants were following stay-at-home advisories and whether they were essential workers in April 2020.

### 2.2.3. Work–Family Conflict

WFC was assessed using adapted versions of two items from the General Social Use Survey (Statistics Canada 2015). The first item assessed FIW conflicts: "How often was it difficult to fulfill your work or studying responsibilities because of your family responsibilities (please include all family responsibilities including homeschooling, childcare, support of your spouse, parents, and other relatives)?" The second item assessed WIF conflicts: "How often was it difficult to fulfill your family responsibilities (including homeschooling[1], childcare, support of your spouse, parents, and other relatives) because of the amount of time you spent on your job or studying?" Response options for each were on a 4-point relative frequency scale from "never" (scored as 1) to "all of the time" (scored as 4), consistent with the scales from the General Social Use Survey (Statistics Canada 2015).

### 2.2.4. Alcohol Use

The Quantity/Frequency/Peak Alcohol Use Index (Dimeff et al. 1999) was used to capture alcohol use. Questions were asked about: number of drinking days (i.e., drinking frequency; possible range = 0–30); number of drinks consumed on a typical drinking day (i.e., typical quantity; range = 0–25); and maximum number of drinks during the heaviest episode (i.e., peak drinks; range = 0–25).

### *2.3. Analysis Plan*

Hypotheses 1 through 2 were evaluated using the entire sample (n = 758 couples) to evaluate how couple-level homeschooling status was related to individuals' WFC and alcohol use. We then included only homeschooling couples (n = 211 couples) to examine effects of time spent homeschooling (Hypotheses 3 through 6). To model the non-independent nature of our data, Actor–Partner Interdependence Models (APIMs; Cook and Kenny 2005) were used. APIMs simultaneously estimate actor effects—the association between one's own predictor (e.g., time spent homeschooling) and one's own outcome—and partner effects, the association between a partner's predictor (e.g., partner time spent homeschooling) and one's own outcome. Models with the WFC indices as outcomes were modeled using traditional APIM estimation; models with alcohol-use outcomes (i.e., models with count and non-normally distributed outcome variables) were analyzed with generalized estimating equations (GEE) APIM models (Loeys et al. 2014) estimating a negative binomial distribution. The interpretation of the coefficient (b) is different depending on the equation used. Whereas in linear regression, coefficients represent the change in the outcome variable for every one-unit increase in the predictor variable (given all other predictors held constant), negative binomial regression models model the log of the expected count as a function of the predictor variables. As such, negative binomial regression coefficients are interpreted as the difference in the logs of expected counts of the outcome variable for every one-unit change in the predictor variable (given all other predictors held constant). It is often preferred to interpret the exponentiated coefficients (i.e., rate ratios; RR) in negative binomial models, and these are therefore also presented in our results. In all models, both partners' ages were included as covariates and all predictors were grand mean centered ( Cook and Kenny 2005).

To be inclusive of all participants regardless of sexual orientation or gender identity, we tested hypotheses regarding homeschooling status/time effects on the outcomes using models that were indistinguishable by gender, with the entire sample (n = 758 couples)

for homeschooling status and all homeschooling couples (n = 211 couples) for time spent homeschooling. Tests of our gender moderation hypotheses were performed using distinguishable dyad models restricted to the mixed-sex couples only (n = 711 couples for tests of homeschooling status; n = 182 couples for tests of time spent homeschooling). Interactions were probed by estimating the simple slopes (i.e., actor and partner effects) of homeschooling status/hours homeschooling on the outcomes separately for men and women.

## 3. Results

Descriptive statistics and bivariate correlations between all study variables are presented in Table 2.

**Table 2.** Descriptive Statistics and Bivariate Correlations among Study Variables.

| | 1 | 2 | 3 | 4 | 5 | 6 | 7 | *Mean (SD) for HS* |
|---|---|---|---|---|---|---|---|---|
| 1. Homeschooling [a] | - | 0.28 *** | 0.26 *** | 0.09 | 0.17 * | 0.06 | 0.10 | 7.22(8.36) |
| 2. FIW | 0.25 *** | - | 0.83 *** | 0.14 ** | 0.17 * | 0.02 | 0.17 * | 1.92(0.91) |
| 3. WIF | 0.27 *** | 0.81 *** | - | 0.14 ** | 0.25 ** | 0.12 | 0.28 *** | 1.84(0.84) |
| 4. Drinking Frequency | −0.03 | −0.02 | 0.01 | - | 0.26 *** | 0.26 *** | 0.34 *** | 6.37(9.15) |
| 5. Drinking Quantity | 0.05 | −0.02 | 0.17 *** | 0.20 *** | - | 0.55 *** | 0.45 *** | 2.62(2.74) |
| 6. Peak Drinking | 0.06 | −0.15 *** | 0.06 | 0.23 *** | 0.50 *** | - | 0.66 *** | 4.17(4.38) |
| 7. Heaviest Drinking Episode | 0.08 * | 0.17 *** | 0.21 *** | 0.29 *** | 0.37 *** | 0.53 *** | - | 1.22(1.95) |
| *Mean (total sample)* | 0.28 | 1.62 | 1.55 | 6.88 | 2.42 | 3.77 | 0.97 | - |
| *SD (total sample)* | 0.45 | 0.83 | 0.77 | 9.37 | 2.51 | 3.90 | 1.90 | - |

*Note.* Correlations for the whole sample are below the diagonal and for the homeschooling-only sample above the diagonal. [a] Below the diagonal, "homeschooling" represents homeschooling status (coded 0 [no] and 1 [yes]) and above the diagonal, "homeschooling" represents hours/week spent homeschooling. * $p < 0.05$; ** $p < 0.01$; *** $p < 0.001$. According to guidelines set in research (Cohen 1988), correlations of $r = 0.10$ are considered small, $r = 0.30$ are considered medium and $r = 0.50$ are considered large.

### 3.1. Gender Differences in WFC

Consistent with H1, women experienced more FIW conflicts than men, $t(479) = −3.68$, $p < 0.001$. Contrary to H1, men did not experience significantly more WIF conflicts than women, $t(467) = −0.93$, $p = 0.354$.

### 3.2. Homeschooling Status

Consistent with H2, homeschooling couples reported more WFC than non-homeschooling couples—both for FIW ($p = 0.003$) and WIF ($p < 0.001$) (Table 3).

Partially consistent with H2, gender significantly moderated the effect of homeschooling status on FIW conflicts ($p = 0.031$) but failed to significantly moderate the effect of homeschooling status on WIF conflicts. In the case of FIW, the effect of homeschooling status was stronger in women ($p < 0.0001$) than men ($p = 0.023$).

Contrary to H2, homeschooling couples did not report greater drinking in April 2020 than non-homeschooling couples on any index (Table 3). And contrary to H2, gender failed to significantly moderate the effect of homeschooling status on any drinking index, indicating no effects of couple-level homeschooling status on any aspect of drinking in either women or men.

**Table 3.** Actor–Partner Interdependence Models (Indistinguishable) Examining Partners' Effects of Homeschooling on Work Family Conflict and Alcohol Use.

| Outcome | Predictor | $b$ [b] | RR | SE($b$) | $Z/t$ [b] | $p$ | 95% LLCI | 95% ULCI |
|---|---|---|---|---|---|---|---|---|
| FIW | Actor age | −0.018 | - | 0.003 | −5.37 *** | <0.001 | −0.025 | −0.012 |
| | Partner age | −0.002 | - | 0.003 | −0.63 | 0.528 | −0.009 | 0.005 |
| | Homeschooling | 0.190 | - | 0.063 | 3.02 ** | 0.003 | 0.067 | 0.314 |
| WIF | Actor age | −0.012 | - | 0.003 | −3.95 *** | <0.001 | −0.019 | −0.006 |
| | Partner age | −0.009 | - | 0.003 | −2.85 ** | 0.005 | −0.015 | −0.003 |
| | Homeschooling | 0.194 | - | 0.057 | 3.41 ** | 0.001 | 0.082 | 0.305 |
| Drinking Frequency | Actor age | 0.019 | 1.019 | 0.005 | 3.82 *** | <0.001 | 0.009 | 0.030 |
| | Partner age | −0.007 | 0.993 | 0.005 | −1.31 | 0.188 | −0.017 | 0.003 |
| | Homeschooling | 0.058 | 1.060 | 0.108 | −0.155 | 0.271 | 0.530 | 0.594 |
| Drinking Quantity | Actor age | 0.005 | 1.005 | 0.004 | 1.18 | 0.239 | −0.003 | 0.013 |
| | Partner age | −0.013 | 0.987 | 0.005 | −2.59 * | 0.010 | −0.023 | −0.003 |
| | Homeschooling | 0.001 | 1.001 | 0.102 | 0.01 | 0.994 | −0.198 | 0.200 |
| Peak Drinking | Actor age | 0.008 | 1.008 | 0.005 | 1.79 | 0.074 | −0.001 | 0.017 |
| | Partner age | −0.014 | 0.986 | 0.005 | −3.03 ** | 0.003 | −0.023 | −0.005 |
| | Homeschooling | 0.032 | 1.033 | 0.104 | 0.31 | 0.076 | −0.171 | 0.235 |

*Note*. Of the six models presented here, the last three (drinking indices) use generalized estimating equations methodology with a negative binomial distribution specified to account for the count and non-normal distribution of the drinking variables. FIW and WIF were normally distributed and as such, traditional linear APIM models were used. [b] The test statistic differs as a function of whether the estimation method is traditional APIM (i.e., $t$) or GEE APIM (i.e., $Z$). * $p < 0.05$; ** $p < 0.01$; *** $p < 0.001$.

### 3.3. Time Spent Homeschooling

Consistent with H3, paired $t$-tests indicated that women reported significantly more time spent homeschooling ($M$ = 8.10 hrs/wk, $SD$ = 7.89) than men ($M$ = 6.53 hrs/wk, $SD$ = 8.84), $t(192)$ = −2.57, $p$ = 0.011.

Consistent with H4, one's own time spent homeschooling was significantly related to one's own WFC in the homeschooling couples—both for FIW ($p$ = 0.001) and WIF ($p$ = 0.017) (Figure 1).

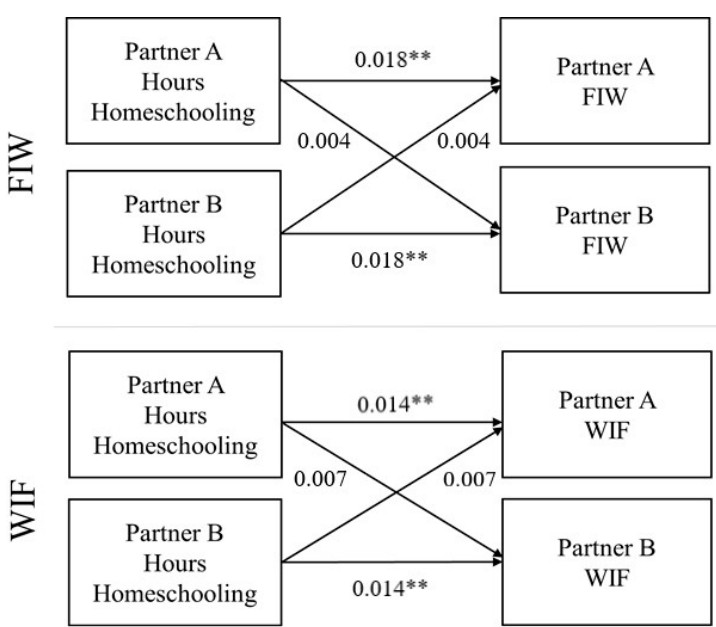

**Figure 1.** Actor–Partner Interdependence Models (APIMs; Indistinguishable Dyads) examining actor and partner effects of time spent homeschooling on family interference with work (FIW) and work interference with family (WIF). * $p < 0.05$; ** $p < 0.01$; *** $p < 0.001$.

However, inconsistent with H6, partner time spent homeschooling was unrelated to one's own work–family conflict—either for FIW conflicts or for WIF conflicts (Figure 1). Neither actor nor partner effects on either index of WFC were significantly moderated by gender, inconsistent with H5 (Table 4). The effect of partner age on WIF was significant ($p = 0.003$).

**Table 4.** Actor–Partner Interdependence Models (Distinguishable Dyads) Examining Partners' Hours spent Homeschooling on Work Family Conflict and Alcohol Use and Interactions with Gender.

| Step | Outcome | Predictor | $b$ [b] | RR | SE($b$) | $Z/t$ [b] | $p$ | 95% LLCI | 95% ULCI |
|---|---|---|---|---|---|---|---|---|---|
| 1 | FIW | Actor age | −0.013 | - | 0.009 | −1.43 | 0.153 | −0.030 | 0.005 |
| | | Partner age | −0.014 | - | 0.009 | −1.64 | 0.103 | −0.031 | 0.003 |
| | | Gender | −0.014 | - | 0.055 | −2.47 * | 0.015 | −0.245 | −0.027 |
| | | Actor homeschooling | 0.015 | - | 0.006 | 2.51 * | 0.013 | 0.003 | 0.027 |
| | | Partner homeschooling | 0.005 | - | 0.006 | 0.87 | 0.386 | −0.006 | 0.017 |
| 2 | | Actor homeschooling x gender | 0.003 | - | 0.007 | 0.44 | 0.658 | −0.011 | 0.017 |
| | | Partner homeschooling x gender | −0.003 | - | 0.007 | −0.45 | 0.651 | −0.017 | 0.011 |
| 1 | WIF | Actor age | 0.003 | - | 0.008 | 0.31 | 0.755 | −0.014 | 0.019 |
| | | Partner age | −0.025 | - | 0.008 | −3.05 ** | 0.003 | −0.041 | −0.009 |
| | | Gender | −0.180 | - | 0.053 | −3.38 ** | 0.001 | −0.285 | −0.075 |
| | | Actor homeschooling | 0.007 | - | 0.006 | 1.21 | 0.227 | −0.005 | 0.019 |
| | | Partner homeschooling | 0.010 | - | 0.006 | 1.79 | 0.075 | −0.001 | 0.024 |
| 2 | | Actor homeschooling x gender | 0.005 | - | 0.007 | 0.80 | 0.426 | −0.008 | 0.019 |
| | | Partner homeschooling x gender | 0.002 | - | 0.007 | 0.28 | 0.776 | −0.012 | 0.016 |
| 1 | Drinking Frequency | Actor age | 0.010 | 1.010 | 0.013 | 0.75 | 0.451 | −0.016 | 0.037 |
| | | Partner age | 0.015 | 1.015 | 0.012 | 1.24 | 0.214 | −0.008 | 0.037 |
| | | Gender | 0.244 | 1.276 | 0.095 | 2.57 * | 0.010 | 0.058 | 0.429 |
| | | Actor homeschooling | 0.020 | 1.020 | 0.010 | 1.99 * | 0.046 | 0.001 | 0.040 |
| | | Partner homeschooling | −0.003 | 0.997 | 0.011 | −0.32 | 0.746 | −0.024 | 0.017 |
| 2 | | Actor homeschooling x gender | −0.029 | 0.971 | 0.011 | −2.61 ** | 0.009 | −0.052 | −0.007 |
| | | Partner homeschooling x gender | 0.038 | 1.039 | 0.011 | 3.34 *** | <0.001 | 0.016 | 0.060 |
| 1 | Drinking Quantity | Actor age | 0.006 | 1.006 | 0.011 | 0.51 | 0.608 | −0.016 | 0.028 |
| | | Partner age | 0.004 | 1.004 | 0.010 | 0.40 | 0.690 | −0.015 | 0.023 |
| | | Gender | 0.101 | 1.106 | 0.078 | 1.30 | 0.194 | −0.051 | 0.253 |
| | | Actor homeschooling | 0.006 | 1.006 | 0.011 | 0.58 | 0.562 | −0.015 | 0.028 |
| | | Partner homeschooling | 0.010 | 1.010 | 0.012 | 0.83 | 0.405 | −0.013 | 0.032 |
| 2 | | Actor homeschooling x gender | −0.001 | 0.999 | 0.011 | −0.05 | 0.956 | −0.023 | 0.022 |
| | | Partner homeschooling x gender | 0.006 | 1.006 | 0.014 | 0.45 | 0.654 | −0.021 | 0.033 |
| 1 | Peak Drinking | Actor age | 0.011 | 1.011 | 0.011 | 0.98 | 0.329 | −0.011 | 0.033 |
| | | Partner age | −0.009 | 0.991 | 0.010 | −0.93 | 0.350 | −0.028 | 0.010 |
| | | Gender | 0.102 | 1.107 | 0.103 | 0.99 | 0.321 | −0.010 | 0.304 |
| | | Actor homeschooling | 0.001 | 1.001 | 0.010 | 0.08 | 0.939 | −0.018 | 0.020 |
| | | Partner homeschooling | 0.015 | 1.015 | 0.011 | 1.45 | 0.147 | −0.005 | 0.036 |
| 2 | | Actor homeschooling x gender | −0.005 | 0.995 | 0.010 | −0.46 | 0.648 | −0.025 | 0.016 |
| | | Partner homeschooling x gender | 0.006 | 1.006 | 0.012 | 0.49 | 0.622 | −0.018 | 0.029 |

*Note.* Of the six models presented here, the last three (drinking indices) use generalized estimating equations (GEE) methodology with a negative binomial distribution specified to account for the count and non-normal distribution of the drinking variables. FIW and WIF were normally distributed and as such, traditional linear APIM models were used. [b] The test statistic differs as a function of whether the estimation method is traditional APIM (i.e., $t$) or GEE APIM (i.e., $Z$). * $p < 0.05$; ** $p < 0.01$; *** $p < 0.001$.

In terms of alcohol use, consistent with H6, partner time spent homeschooling was significantly positively related to one's own drinking frequency ($p = 0.012$), quantity ($p = 0.030$), and peak drinking ($p = 0.018$)—suggesting adverse cross-over effects (Figure 2). Inconsistent with H4, one's own time spent homeschooling was unrelated to one's own drinking on any of the drinking indices in the indistinguishable dyad analysis (Figure 2). However, in the distinguishable (by gender) dyad analyses, both the actor ($p = 0.009$) and partner ($p < 0.001$) effects of time spent homeschooling on drinking frequency were significantly moderated by gender (Table 4).

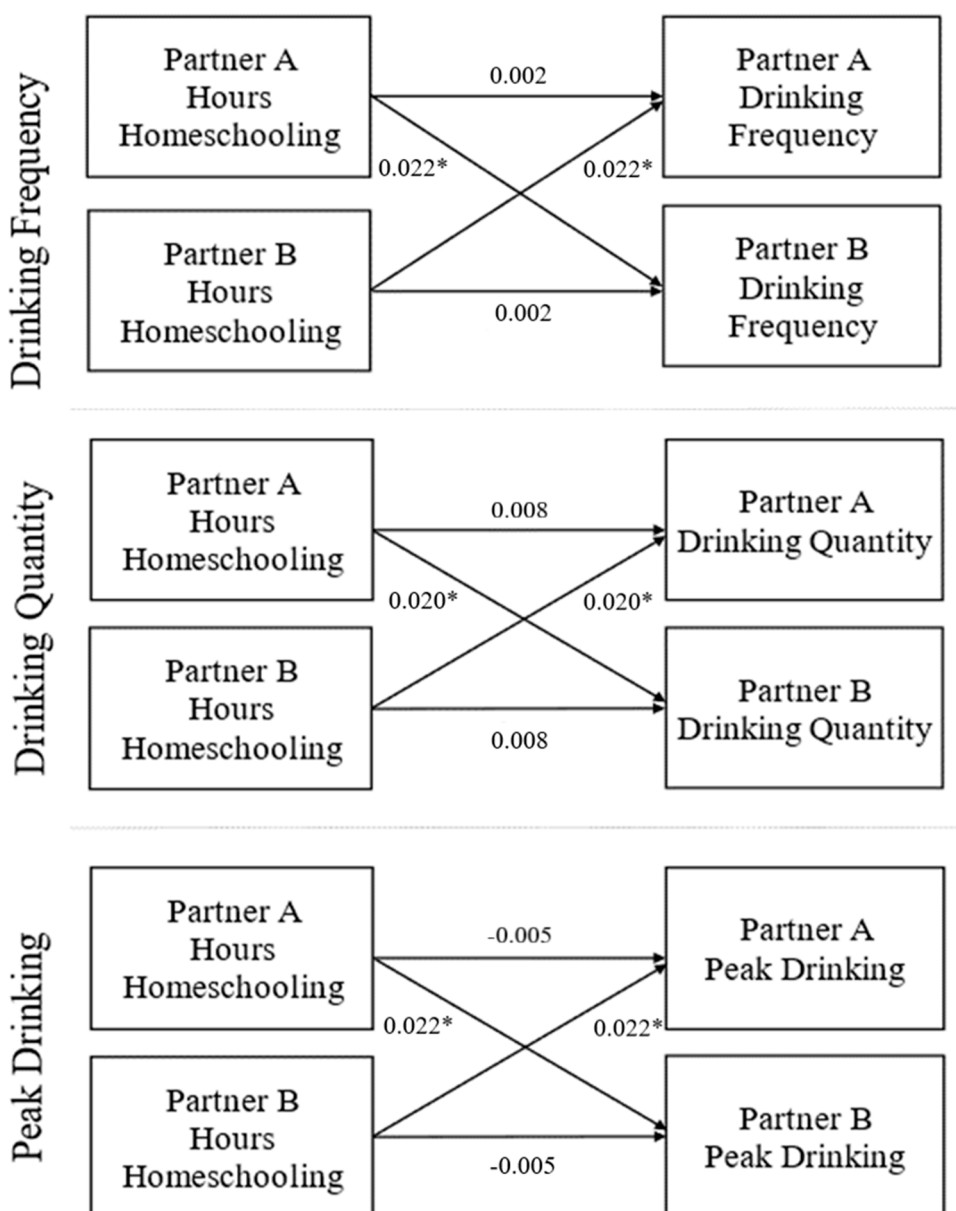

**Figure 2.** Actor–Partner Interdependence Model (APIM; Indistinguishable Dyads) examining actor and partner effects of time spent homeschooling on drinking frequency, quantity, and COVID-19 peak drinking. * $p < 0.05$; ** $p < 0.01$; *** $p < 0.001$.

Consistent with H5 and H6, respectively, the woman's time spent homeschooling was significantly and positively related to both her own ($p = 0.011$) and her partner's ($p = 0.022$) drinking frequency, consistent with both the multiple-burdens hypothesis and adverse cross-over effects, respectively (Figure 3). In contrast, the man's time spent homeschooling was unrelated to his own drinking frequency. Additionally, the man's time spent homeschooling was significantly *negatively* related to his partner's drinking frequency ($p = 0.015$), suggesting *protective* cross-over effects of the man's time spent homeschooling on the woman's drinking (Figure 3).

Supplementary analyses were run to test effects with only the participants who indicated they were mandatorily homeschooling due to the pandemic. Results were unchanged with this subsample with the exception that the significant protective effect of males' hours spent homeschooling on female drinking frequency became nonsignificant ($p = 0.134$), likely due to the reduced power given the smaller homeschooling sample size.

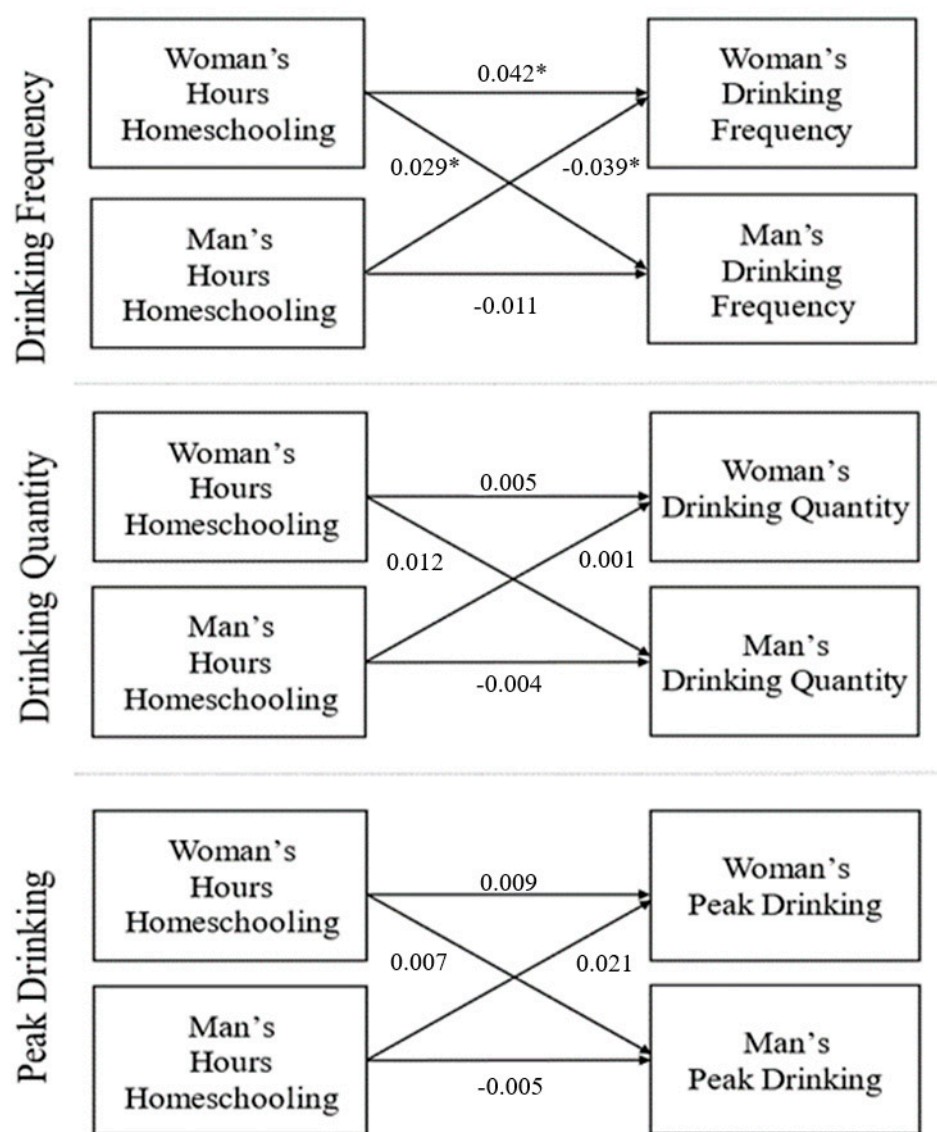

**Figure 3.** Actor–Partner Interdependence Models (APIM; Distinguishable Dyads) examining hours spent homeschooling on drinking outcomes for men and women. * $p < 0.05$; ** $p < 0.01$; *** $p < 0.001$.

## 4. Discussion

In the present study, we examined COVID-19 mandated homeschooling as a potential risk factor for increased WFC and alcohol use in couples during stay-at-home advisories and tested whether these effects varied by gender. We were interested in doing so because of the repeated findings showing parents are at particularly increased risk for decreased mental health due to the COVID-19 pandemic and its containment measures (Brooks et al. 2020; Wenham et al. 2020) and given pre-pandemic findings of stronger links of WFC to drinking in women (Kuntsche and Kuntsche 2021; Kuntsche et al. 2009; Roos et al. 2006). Further, we explored if there were cross-over effects of mandated homeschooling in couples (Neff and Karney 2007), and whether these were different for women and men.

In terms of WFC, we found gender differences in FIW but not WIF conflicts. Consistent with previous research (Fu and Shaffer 2001), women experienced more FIW conflicts than men. Interestingly, however, men did not experience more WIF conflicts than women. This is inconsistent with pre-pandemic studies showing men are more likely to struggle with work interferences at home (Kulik et al. 2016). A potential explanation is that, during COVID-19, men were more likely to be working from home and/or spending more time at home. This might have allowed them to be better able to balance their work and

family responsibilities than in pre-pandemic contexts. Additionally, many unsupportive workplace policies (i.e., inflexible scheduling and lack of paid family leave) that are likely to foster more WIF were eliminated by COVID-19 containment measures (Pedulla and Thébaud 2015).

Further, couples who were homeschooling one or more children in Grades 1–12 during the lockdown in April 2020 experienced more WFC compared to couples who were not homeschooling children. This is consistent with the multiple burdens hypothesis (Kuntsche et al. 2009), in that couples who held the additional role of homeschooling a child showed more WFC. Moreover, for those couples who were homeschooling, greater time spent homeschooling was associated with greater WFC. Our findings build on data showing that Canadian adults with children between the ages of 6 and 12 showed the highest levels of WFC in April 2020 (Schieman et al. 2021), by clarifying that these effects are due to the context of mandated homeschooling.

Interestingly, in this data set, the requirements of mandatory homeschooling created WFC in terms of both homeschooling interfering with work (FIW) and parents' work requirements (including working from home) interfering with the ability to provide the homeschooling their children needed (WIF). We also observed that having a younger partner was associated with an adverse effect on WIF. One potential explanation is that younger partners might provide less support during a stressful time such as a global pandemic and thereby contribute to greater WIF. The reasons for this effect remain to be determined. Further, given that women take on more childcare and educational responsibilities than men both pre-pandemic (Ferrant et al. 2014; Keene and Quadagno 2004) and during the pandemic (Calarco et al. 2020; Del Boca et al. 2020; Miller 2020; Shafer et al. 2020), we anticipated that this unequal burden would lead to greater effects of homeschooling status on WFC for women than men. This hypothesis was supported, at least for FIW conflicts. While both men and women experienced effects of homeschooling status on FIW, the impact was greater for women. The latter suggests that the different levels of WFC in homeschooling versus non-homeschooling couples is driven by homeschooling (accounted for by family duties) and not by the observed difference between the groups in full-time work. Further, consistent with research showing family demands are a more important predictor of WFC for women than for men (Elliott 2008; McElwain et al. 2005), the present study showed that the additional familial duty of COVID-19 mandated homeschooling is a particularly important risk factor for increased WFC in women. Moreover, given that this statistical moderation effect of gender was specific to FIW conflicts as opposed to WIF conflicts, this suggests that the increased WFC experienced by women involved in mandatory homeschooling is specifically due to the increased demands of family homeschooling responsibilities interfering with their work duties.

This increased WFC among women homeschoolers may play into their decisions to return to the workforce following the pandemic and thereby contribute to the predicted pandemic-related "she-cession" (Alini 2020), with greater adverse economic impacts of the pandemic expected for women than for men.

In terms of alcohol use, analyses by gender suggested that the multiple-burdens hypothesis might need fine-tuning. Overall, effects on drinking behavior in the full sample were not consistent with the multiple-burdens hypothesis (Kuntsche et al. 2009), with no significant actor effects of one's own time spent homeschooling on any of the drinking indices in the total sample. However, we did observe the expected gender moderating effect for drinking frequency where there was a significant actor effect, but only among women. This suggests the multiple-burdens hypothesis may specifically explain the effects of homeschooling on more frequent drinking among women. This is consistent with pre-pandemic findings suggesting stronger links of WFC with drinking behavior in women than in men (Kuntsche and Kuntsche 2021; Kuntsche et al. 2009; Roos et al. 2006). Greater links of WFC to alcohol use in women than in men might be due to increased negative affect and psychological distress (Frone et al. 1994), which are both linked to increased alcohol use in women (Kuntsche et al. 2015). Our findings are also consistent with recent

work showing pandemic-related psychological distress is more strongly positively related to drinking in women than in men (Rodriguez et al. 2020).

For those homeschooling, it was revealed that women reported more hours spent homeschooling than men. It is possible that this difference is minimized relative to reality given that these hours came from men's and women's self-reports of their own behavior (Shafer et al. 2020). Our findings are in line with recent data showing that women are more likely to be taking on the role of "teacher" than men during the pandemic (Del Boca et al. 2020; Miller 2020; Shafer et al. 2020). Pre-pandemic evidence on voluntary homeschooling suggests that this additional burden takes a toll on the mental health of mothers (Baker 2019; Lois 2006), a finding that has been replicated in COVID-19 pandemic research (APA 2020; Schmidt et al. 2020). It is concerning that greater involvement in mandatory homeschooling during the pandemic is linked with increased drinking frequency in women, given that increased drinking may interfere with the ability to parent through this challenging time. Indeed, even non-dependent parental drinking has been associated with children's experiences of negative outcomes following their parent's drinking (Bryant et al. 2020).

Considering these relations within couples yielded important findings regarding the couple dynamic. Consistent with our hypotheses, time spent homeschooling by the partner was associated with increased drinking by the individual on all three drinking indices (quantity, frequency, and peak), pointing to adverse cross-over effects of a partner's homeschooling involvement on the individual's drinking behavior. Given our findings with WFC, one possibility is that the individual observes and is emotionally affected by their homeschooling partner's increased stress, with the individual then using alcohol to cope with this "contagious" stress (Neff and Karney 2007). We also observed gender moderation of the adverse effects of partner time spent homeschooling on own drinking in the case of drinking frequency. Specifically, when the woman engaged in more hours of homeschooling, the drinking frequency of the male partner increased. This is consistent with past research showing that partner cross-over effects of stress are significantly stronger for husband than wife actors (Neff and Karney 2007). Thus, men may be more susceptible to increased drinking in response to the hypothesized social contagion effects of the WFC-related stress imposed by mandatory homeschooling. The latter is worrisome given that fathers' increased drinking could put homeschooling mothers at risk for escalating conflict and domestic violence. A large body of research links alcohol use in men to both romantic conflict (Capaldi et al. 2012) and to intimate partner violence (IPV) against women (Foran and O'Leary 2008). Our findings might support the predictions that COVID-19 could increase relationship conflict (Pietromonaco and Overall 2020). Indeed, a recent study found pandemic-related increases in mothers' frustrations with their partners (Calarco et al. 2020). Further, a systematic review of studies examining the association between the COVID-19 pandemic and IPV confirmed a large, global increase in the number of IPV reports (Landoni and Ionio 2020). Accordingly, the World Health Organization (2020) recently released data confirming a global increase in reports of IPV since the onset of the pandemic. In Canada, between March and April 2020, 1 out of 10 Canadian women reported being very or extremely concerned about the possibility of violence in the home during the pandemic (Statistics Canada 2020). Our findings point to the possible dynamic role of mandatory homeschooling related stress in the mother triggering increased drinking in the father, which further increases the risk of romantic conflict and IPV perpetration during the pandemic.

Another interesting result of looking at these effects within couples was that this lens revealed a gender-specific protective partner effect of homeschooling on drinking frequency. In contrast to our observations with women's homeschooling, when the man engaged more in homeschooling, taking on the role of teacher, the drinking frequency of the woman partner *decreased*. While overall, the responsibility of homeschooling during the pandemic fell to women (Del Boca et al. 2020; Miller 2020; Shafer et al. 2020), our data show that when men took a greater share of this responsibility (as assessed through increased time spent homeschooling), drinking frequency in women decreased. This shows

a protective cross-over effect, with women's drinking during the pandemic tempered when men are more invested in homeschooling. This falls in line with evidence suggesting that more equitable division of labor in mixed-sex couples has a protective effect on mental health (Kalmijn and Monden 2012).

*4.1. Limitations and Future Directions*

There are some limitations to our study that should be considered. While we have determined the statistical significance of these findings, more work is needed to establish their practical significance. For instance, future work could ask parents what degree of change in drinking and/or in WFC is noticeable and disruptive. Further, the homeschooling literature has predominantly focused on voluntary contexts (Guterman and Neuman 2018, 2020). There may be both important similarities and differences between parents who are homeschooling on a mandated versus voluntary basis. Unfortunately, the sample size of this data set was not large enough to explore such effects directly. Given that voluntary homeschooling is likely to have become more stressful during the COVID-19 pandemic, with reduced access to supports on which these parents may normally rely (e.g., libraries and other homeschooling families), we felt justified in combining mandated homeschoolers and voluntary homeschoolers in our sample. Future research should examine the differences between parents who choose to school their children at home versus those who are obligated due to pandemic restrictions. Moreover, the current study depended on retrospective reports, which introduced a delay that inevitably increased the likelihood of error. Nonetheless, capturing the experiences of parents specifically in April 2020, during unexpected nationwide school closures, is vital even some time later. Future studies should look at whether the effects of homeschooling on WFC and alcohol use persist at later periods in time during the COVID-19 pandemic. Further, our cross-sectional analyses only allow for the examination of associations. Longitudinal data would permit temporally separating associations and allow for more confidence in the directionality of effects. Moreover, in our sample, homeschooling couples and non-homeschooling couples differed on more than age. Future studies should try to recruit more comparable groups. In addition, this sample was entirely Canadian, with most participants identifying as White and being university-educated; thus, there may be limitations to the generalizability of these results. Future research should explore the effects of mandated homeschooling on WFC and drinking within couples from different ethnic backgrounds and socioeconomic statuses, given that COVID-19 is predicted to exacerbate social inequalities (Cénat et al. 2020). In addition, it is important to recognize that mandated homeschooling might take a particular toll on parents of children who are likely to experience more difficulties with remote learning, such as children diagnosed with learning disabilities or attention deficit hyperactivity disorder (Becker et al. 2020; Cataudella et al. 2021; Sibley et al. 2021). Accordingly, this should be explored in future research. Further, future work should look at the differential impact of mandated homeschooling status on WFC and drinking in same-sex versus different-sex couples. Though our non-gender specific analyses were inclusive as they included same-sex couples, we did not have a sufficient sample size of same-sex couples to make comparisons between opposite sex and same-sex couples. Future studies should focus on specific recruitment of same-sex couples to achieve required sample sizes. We also recognize that we were not able to capture the amount of support that parents were receiving from their child's school. As such, more work is needed to address how different curriculums (mandatory versus non-mandatory, different time requirements, etc.) impact WFC and alcohol use behaviours in parents. Lastly, our study included homeschooling as part of our WFC measures. Though this reflected the reality of family responsibilities in Canada in April 2020, it might have led to measurement redundancy between the homeschooling predictor and WFC outcome. Future research should examine WFC for each different type of family responsibility separately (i.e., homeschooling, childcare, and support of spouse, parents, and other relatives).

### 4.2. Implications and Applications

Our study is novel in that it is the first to examine the differential impact of home-schooling on WFC and alcohol use behaviour in couples in the context of the COVID-19 pandemic. We found that removing parenting supports by closing schools during the pandemic has substantial negative associations with both work–family conflict and drinking, highlighting the importance of better assisting families through current and future large-scale viral outbreaks and other crises requiring school closures. First, our results build on previous research by extending the finding that women experience more FIW than men to the context of the COVID-19 pandemic. Given that these conflicts are found to be harmful to the mental health of women (Kulik and Liberman 2013), especially compounded with the uncertainty and anxiety about the pandemic, more effective supports for women are needed during these unprecedented times. Second, consistent with concerns about the mental health effects of public health responses to the pandemic (Asmundson et al. 2020) our findings provide further support that closing schools and requiring parents to homeschool has significant adverse consequences, in this case increasing WFC and drinking. These findings support recommendations from a recent policy brief by the Royal Society of Canada (Asmundson et al. 2020) that provinces and territories should make efforts to keep children in open schools and consider whether the cost of closing schools outweighs its benefits (Deacon et al. forthcoming). Indeed, data suggest that the strategy of closing schools has a much smaller impact on disease spread relative to other public health measures (UNICEF 2020) and modelling suggests it is associated with lower years of life loss than school closures (Christakis et al. 2020). Third, should school closures be required, our findings suggest that more educational supports should be provided to parents to reduce the extra role strain placed on them by having to assume the role of teacher. Fourth, more mental health supports should be allocated for couples having to homeschool during the pandemic given their increased WFC and alcohol use. Accordingly, our data are consistent with recommendations that parents be better supported when educating children at home due to the pandemic, including increased mental health and substance-use services (Deacon et al. forthcoming). Finally, more egalitarian division of homeschooling labor between couple members is needed to prevent adverse effects on both individuals. Indeed, our study is the first to show that men assuming a fairer share of the homeschooling burden can even have protective effects on their female partner's coping behaviors during a global crisis.

**Author Contributions:** Conceptualization, D.I.D., S.H.D., L.M.R. and S.H.S.; Data curation, M.M.E. and F.E.K.; Formal analysis, L.M.R.; Funding acquisition, S.H.D., L.M.R., R.N.-A., S.M., A.A. and S.H.S.; Investigation, D.I.D., S.H.D., L.M.R. and S.H.S.; Methodology, S.H.D., L.M.R. and S.H.S.; Project administration, D.I.D., M.M.E. and F.E.K.; Resources, S.H.D., L.M.R. and S.H.S.; Supervision, S.H.S.; Validation, M.M.E. and F.E.K.; Visualization, L.M.R.; Writing—original draft, D.I.D.; Writing—review & editing, D.I.D., S.H.D., L.M.R., S.B.S., R.N.-A., M.M.E., S.M., A.A., F.E.K. and S.H.S. All authors have read and agreed to the published version of the manuscript.

**Funding:** This research was funded by a special COVID-19 research grant from the Dalhousie University Department of Psychiatry Research Fund as well as the Social Sciences and Humanities Research Council of Canada (SSHRC) Partnership Engage Grant, grant number 1008-2020-0220.

**Institutional Review Board Statement:** The study was conducted according to the guidelines of the Declaration of Helsinki, and approved by the Research Ethics Board of Dalhousie University (#2020-5166, 4 June 2020).

**Informed Consent Statement:** Informed consent was obtained from all participants involved in the study.

**Data Availability Statement:** Not applicable.

**Conflicts of Interest:** The authors declare no conflict of interest. The funders had no role in the design of the study; in the collection, analyses, or interpretation of data; in the writing of the manuscript; or in the decision to publish the results.

## Notes

[1] The General Social Use Survey was adapted by adding homeschooling to the list of family responsibilities in each WFC item.

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
