# Peer review of "Homeschooling during COVID-19: Gender Differences in Work–Family Conflict and Alcohol Use Behaviour among Romantic Couples"

_socsci, doi:10.3390/socsci10070240_

Round 1

Reviewer 1 Report

REVIEW                                                                                                                                                                May 28, 2021

Social Sciences

Manuscript ID: socsci-1233981

Title: “Mandatory Homeschooling During the COVID-19 Pandemic: Study of Gender Differences in Work-Family Conflict and Alcohol Use Behaviour Among Romantic Couples”

Abstract: COVID-19 forced many parents to do homeschooling. How this may have affected parents’ mutual relationships, workloads and the use of alcohol use are the central questions of the paper. A total of 211 homeschooling couples were compared to 547 non-homeschooling parents in April 2020. Most results were consistent with the existing literature, while a few results deviated from the expected.

Overall Impression: The research focused on an important present topic, was thoroughly researched with mindful insights. The biggest limitation of the study was its time-period, April 2020. That was the first full month of the Covid, and it seems quite possible, if not even highly likely, that the results for later periods could be quite different.

Comments by line:

1-4. The title is far too long and confusing.

34-35. I’m not familiar with the term “romantic couples.” Is that a standard phrase in the literature?

58-59. If homeschool becomes so quickly overwhelming for parents, why are nearly 2 million U.S. children homeschooled?

70-71. I’m somewhat confused. Were the Covid-interviews done in April and June 2020, or only in April? If only in June, it seems quite possible that the lapsed time has already altered people’s answers.

94- It would have been interesting to read some quantitative results from previous studies about drinking, etc. Now everything was reported just quantitatively significant. Yet, many things are statistically significant without any real-life significance.

Table 1. It would warrant a comment that the two groups studied exhibited some considerable differences in their composition: Relationship length, employment status, education, and ethnicity. The geographic coverage seemed about right.

  1. It was hard to tell from the summary statistics table how many couples were close to the three-month threshold. You are certainly right that the length of relationship is an important variable to study.

197-198. I understand why use it, but the 4-point scale is certainly a very rough way to measure things.

205- You present several interesting hypotheses. They are, however, hard to read from the middle of the text. Maybe you could create a list of them?

Table 2. I get back to my earlier point: statistical significance does not automatically mean economic significance. I have been schooled to accept correlations >0.70 as strong. With that criterion, only one pair passes the test. Further, correlations <0.30 are usually to be ignored. The “stars” make everything make most relationship matter a lot. Yet, I think only few actually are (high or no correlation). Maybe you could make a short note of that somewhere in the text?

  1. Women spend significantly more time on homeschooling than men, at least statistically. The statement is correct, of course, but the difference is not that much real terms: 8.10hrs/wk vs 6.53hrs/wk, or +12%. I think much less than most people would have expected.

Figure 1. If I understood correctly the meaning of the APIM numbers, they are were small in real-life terms. If that is not the case, please interpret them more clearly.

Figure 2. Ibid.

Figure 3. Ibid.

  1. Under limitations it would be good to add a few words on the difference between correlation- and regression-based analysis. I can easily come for ten reasons each for why couples may have drunk or fought more in April 2020, but the correlational analysis won’t be able to tell me that.

Finally, in limitations you state quite nicely that while the study is not perfect, it does provide us some new information about a hugely important issue to our society. I totally agree.

+++ Thanks for the well-written paper. It was fun to read it! +++

  1. It wasn’t your research question, but now I just keep wondering and wondering whether the mandatory & non-mandatory curriculum in different provinces would have affected your results?

Author Response

Dear Reviewer,

Thank you for your feedback on our manuscript entitled Mandatory Homeschooling During the COVID-19 Pandemic: A Study of Gender Differences in Work-Family Conflict and Alcohol Use Behaviour Among Romantic Couples. We appreciate the comments provided; we believe it resulted in a stronger and more focused manuscript.

I am taking responsibility for the changes to this manuscript on behalf of my co-authors.

Sincerely,

The Authors

Reviewer 2 Report

This is a significant manuscript that discusses the effects of homeschooling on Work-Family Conflict and Alcohol Use Behaviour under the COVID-19 pandemic.

Although the hypotheses were tested through detailed analysis with reference to literatures, the following points need to be improved.

1, Title

Although the title is "Mandatory schooling," it actually includes voluntary schooling. If this title is to be used, the target population should be limited to the 86% 173 couples of the participants in this analysis. This is especially true if the results of the analysis remain unchanged. However, if it is difficult to make the change due to time constraints (i.e., time for reanalysis or the desire to report the impact of COVID-19 quickly), the title should remove the word Mandatory.

2,Abstract

(1) Duplicated sentences should be avoided.

L10: There were stronger effects on family-interference with work in women

L14-15: Increased work-family conflict in homeschooling couples, especially the stronger effects of homeschooling on work-family conflict in women

L19-21: “Finally, the protective partner effects of men's homeschooling hours on women's drinking frequency suggests that a more egalitarian division of mandatory homeschooling labor may have protective cross-over effects.”

This effect was exceptionally insignificant when the target population was limited to Mandatory schooling (L308-312). Therefore, this result is not applicable to mandatory schooling under COVID-19. There is a discrepancy with the results of the main analysis of the paper.

In other words, if the increase in home schooling time in men does not affect the decrease in drinking frequency in women, that would be the influence of COVID-19 pandemic on mental health in women, and this point should be discussed.

  1. Analysis

As commented to Title, the participants of this study include voluntary schooling. If this title is to be used, the target population should be limited to 86%, 173 couples in this analysis ideally.APIMs were adequately analyzed to examine the hypothesis. However, there were too many hypotheses and should be limited to three or four to focus the issues.

Age is a possible confounder; In Table 3, the percentage of each age group should be shown. In Table 4, partner age is significant, but this point is not discussed in this manuscript and should be added to discussion.

Overall, there are too many hypotheses, which gives the impression of complexity. It is desirable to focus the discussion on gender and alcohol consumption.

Author Response

(The authors gave the same response as above.)

Reviewer 3 Report

The study is well presented and very interesting to read, as well as to understand. What I can say is that although it is indicated in the limitations, given that the study has similar extent with most of the previous studies, it could have been extended a little much beyond the extension in presented. For instances the differences between same sex, and different sex couples could have been an interesting findings within this study's scope. Nevertheless, it is an interesting paper. 

Author Response

Dear Reviewer,

Thank you for your feedback concerning the status of our manuscript entitled Mandatory Homeschooling During the COVID-19 Pandemic: A Study of Gender Differences in Work-Family Conflict and Alcohol Use Behaviour Among Romantic Couples. We appreciate the feedback provided; we believe it resulted in a stronger and more focused manuscript.

I am taking responsibility for the changes to this manuscript on behalf of my co-authors.

Sincerely,

The Authors